# SegNeXt: Rethinking Convolutional Attention Design for Semantic Segmentation

**Meng-Hao Guo**[1]   **Cheng-Ze Lu**[2]   **Qibin Hou**[2]   **Zheng-Ning Liu**[3]
**Ming-Ming Cheng**[2]   **Shi-Min Hu**[1]*

[1]BNRist, Department of Computer Science and Technology, Tsinghua University
[2]TMCC, CS, Nankai University      [3]Fitten Tech, Beijing, China

## Abstract

We present SegNeXt, a simple convolutional network architecture for semantic segmentation. Recent transformer-based models have dominated the field of semantic segmentation due to the efficiency of self-attention in encoding spatial information. In this paper, we show that convolutional attention is a more efficient and effective way to encode contextual information than the self-attention mechanism in transformers. By re-examining the characteristics owned by successful segmentation models, we discover several key components leading to the performance improvement of segmentation models. This motivates us to design a novel convolutional attention network that uses cheap convolutional operations. Without bells and whistles, our SegNeXt significantly improves the performance of previous state-of-the-art methods on popular benchmarks, including ADE20K, Cityscapes, COCO-Stuff, Pascal VOC, Pascal Context, and iSAID. Notably, SegNeXt outperforms EfficientNet-L2 w/ NAS-FPN and achieves $90.6\%$ mIoU on the Pascal VOC 2012 test leaderboard using only $\frac{1}{10}$ parameters of it. On average, SegNeXt achieves about $2.0\%$ mIoU improvements compared to the state-of-the-art methods on the ADE20K datasets with the same or fewer computations.

## 1   Introduction

As one of the most fundamental research topics in computer vision, semantic segmentation, which aims at assigning each pixel a semantic category, has attracted great attention over the past decade. From early CNN-based models, typified by FCN [60] and DeepLab series [4, 5, 7], to recent transformer-based methods, represented by SETR [108] and SegFormer [90], semantic segmentation models have experienced significant revolution in terms of network architectures.

Table 1: Properties we observe from the successful semantic segmentation methods that are beneficial to the boost of model performance. Here, $n$ refers to the number of pixels or tokens. Strong encoder denotes strong backbones, like ViT [16] and VAN [25].

| Properties | DeepLabV3+ | HRNet | SETR | SegFormer | SegNeXt |
|---|---|---|---|---|---|
| Strong encoder | ✗ | ✗ | ✓ | ✓ | ✓ |
| Multi-scale interaction | ✓ | ✓ | ✗ | ✗ | ✓ |
| Spatial attention | ✗ | ✗ | ✓ | ✓ | ✓ |
| Computational complexity | $\mathcal{O}(n)$ | $\mathcal{O}(n)$ | $\mathcal{O}(n^2)$ | $\mathcal{O}(n^2)$ | $\mathcal{O}(n)$ |

---

*S.-M. Hu is the corresponding author. Project page: https://github.com/Jittor/JSeg

36th Conference on Neural Information Processing Systems (NeurIPS 2022).

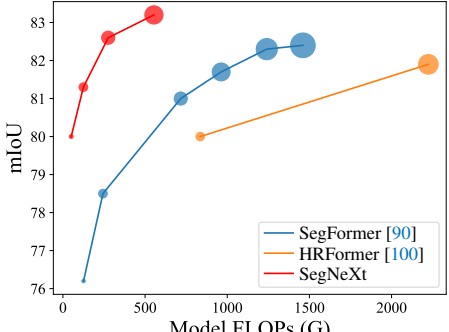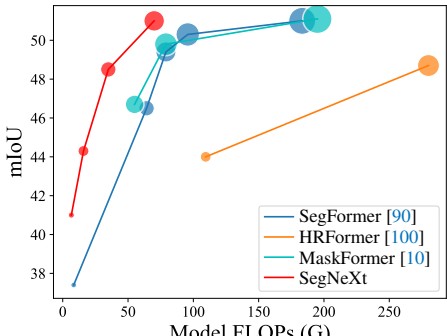

Figure 1: Performance-Computing curves on the Cityscapes (left) and ADE20K (right) validation sets. FLOPs are calculated using an input size of $2,048 \times 1,024$ for Cityscapes and $512 \times 512$ for ADE20K. The size of the circle indicates the number of parameters. Larger circles mean more parameters. We can see that our SegNeXt achieves the best trade-off between segmentation performance and computational complexity.

By revisiting previous successful semantic segmentation works, we summarize several key properties different models possess as shown in Tab. 1. Based on the above observation, we argue a successful semantic segmentation model should have the following characteristics: (i) A strong backbone network as encoder. Compared to previous CNN-based models, the performance improvement of transformer-based models is mostly from a stronger backbone network. (ii) Multi-scale information interaction. Different from the image classification task that mostly identifies a single object, semantic segmentation is a dense prediction task and hence needs to process objects of varying sizes in a single image. (iii) Spatial attention. Spatial attention allows models to perform segmentation through prioritization of areas within the semantic regions. (iv) Low computational complexity. This is especially crucial when dealing with high-resolution images from remote sensing and urban scenes.

Taking the aforementioned analysis into account, in this paper, we rethink the design of convolutional attention and propose an efficient yet effective architecture for semantic segmentation. Unlike previous transformer-based models that use convolutions in decoders as feature refiners, our method inverts the transformer-convolution encoder-decoder architecture. Specifically, for each block in our encoder, we renovate the design of conventional convolutional blocks and utilize multi-scale convolutional features to evoke spatial attention via a simple element-wise multiplication following [25]. We found such a simple way to build spatial attention is more efficient than both the standard convolutions and self-attention in spatial information encoding. For decoder, we collect multi-level features from different stages and use Hamburger [22] to further extract global context. Under this setting, our method can obtain multi-scale context from local to global, achieve adaptability in spatial and channel dimensions, and aggregate information from low to high levels.

Our network, termed SegNeXt, is mostly composed of convolutional operations except the decoder part, which contains a decomposition-based Hamburger module [22] (Ham) for global information extraction. This makes our SegNeXt much more efficient than previous segmentation methods that heavily rely on transformers. As shown in Fig. 1, SegNeXt outperforms recent transformer-based methods significantly. In particular, our SegNeXt-S outperforms SegFormer-B2 (81.3% vs. 81.0%) using only about ⅙ (124.6G vs. 717.1G) computational cost and ½ parameters (13.9M vs. 27.6M) when dealing with high-resolution urban scenes from the Cityscapes dataset.

Our contributions can be summarized as follows:

- We identify the characteristics that a good semantic segmentation model should own and present a novel tailored network architecture, termed SegNeXt, that evokes spatial attention via multi-scale convolutional features.

- We show that an encoder with simple and cheap convolutions can still perform better than vision transformers, especially when processing object details, while it requires much less computational cost.

- Our method improves the performance of state-of-the-art semantic segmentation methods by a large margin on various segmentation benchmarks, including ADE20K, Cityscapes, COCO-Stuff, Pascal VOC, Pascal Context, and iSAID.

## 2 Related Work

### 2.1 Semantic Segmentation

Semantic segmentation is a fundamental computer vision task. Since FCN [60] was proposed, convolutional neural networks (CNNs) [1, 71, 98, 106, 20, 99, 79, 21, 51] have achieved great success and become a popular architecture for semantic segmentation. Recently, transformer-based methods [108, 90, 100, 73, 70, 50, 10, 9] have shown great potentials and outperform CNN-based methods.

In the era of deep learning, the architecture of segmentation models can be roughly divided into two parts: encoder and decoder. For the encoder, researchers usually adopt popular classification networks (*e.g.,* ResNet [28], ResNeXt [91] and DenseNet [33]) instead of tailored architecture. However, semantic segmentation is a kind of dense prediction task, which is different from image classification. The improvement in classification may not appear in the challenging segmentation task [29]. Thus, some tailored encoders appear, including Res2Net [21], HRNet [79], SETR [108], SegFormer [90], HRFormer [100], MPViT [44], DPT [70], *etc*. For the decoder, it is often used in cooperating with encoders to achieve better results. There are different types of decoders for different goals, including achieving multi-scale receptive fields [106, 6, 88], collecting multi-scale semantics [71, 90, 7], enlarging receptive field [4, 4, 69], strengthening edge features [107, 2, 15, 48, 102], and capturing global context [20, 35, 101, 46, 24, 27, 103].

In this paper, we summarize the characteristics of those successful models designed for semantic segmentation and present a CNN-based model, named SegNeXt. The most related work to our paper, is [69], which decomposes a $k \times k$ convolution into a pair of $k \times 1$ and $1 \times k$ convolutions. Though this work has shown large convolutional kernels matter in semantic segmentation, it ignores the importance of multi-scale receptive field and does not consider how to leverage these multi-scale features extracted by large kernels for segmentation in the form of attention.

### 2.2 Multi-Scale Networks

Designing multi-scale network is one of the popular directions in computer vision. For segmentation models, multi-scale blocks appear in both the encoder [79, 21, 75] and the decoder [106, 98, 5] parts. GoogleNet [75] is one of the most related multi-scale architectures to our method, which uses a multi-branch structure to achieve multi-scale feature extraction. Another work that is related to our method is HRNet [79]. In the deeper stages, HRNet also keeps high-resolution features, which are aggregated with low-resolution features, to enable multi-scale feature extraction.

Different from previous methods, SegNeXt, besides capturing multi-scale features in encoder, introduces an efficient attention mechanism and employs cheaper and larger kernel convolutions. These enable our model to achieve higher performance than the aforementioned segmentation methods.

### 2.3 Attention Mechanisms

Attention mechanism is a kind of adaptive selection process, which aims to make the network focus on the important part. Generally speaking, it can be divided into two categories in semantic segmentation [26], including channel attention and spatial attention. Different types of attentions play different roles. For instance, spatial attentions mainly care about the important spatial regions [16, 13, 64, 58, 23]. Differently, the goal of using channel attention is to make the network selectively attend to those important objects, which has been demonstrated important in previous works [31, 8, 80]. Speaking of the recent popular vision transformers [16, 58, 94, 81, 82, 57, 90, 34, 56, 100, 93], they usually ignore adaptability in channel dimension.

Visual attention network (VAN) [25] is the most related work to SegNeXt, which also proposes to leverage the large-kernel attention (LKA) mechanism to build both channel and spatial attention. Though VAN has achieved great performance in image classification, it neglects the role of multi-scale feature aggregation during the network design, which is crucial for segmentation-like tasks.

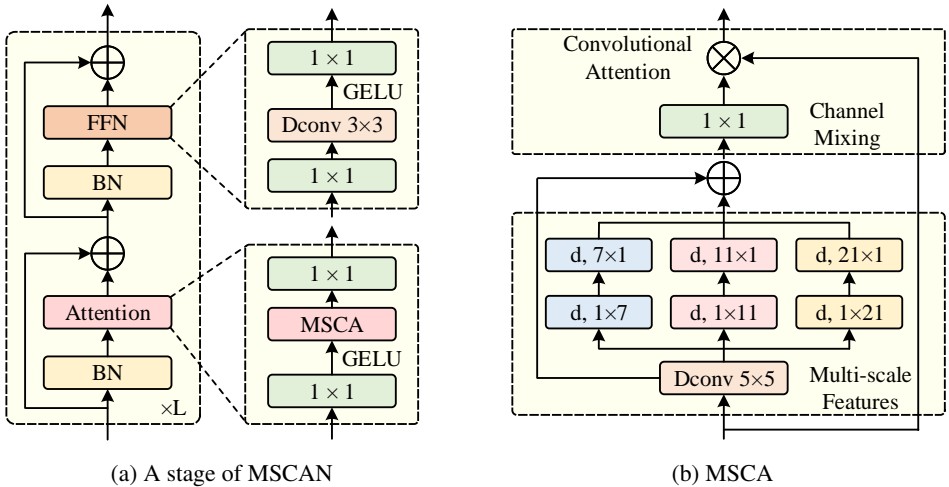

|                       |                       |
| :-------------------: | :-------------------: |
| (a) A stage of MSCAN  |       (b) MSCA        |

Figure 2: Illustration of the proposed MSCA and MSCAN. Here, $d, k_1 \times k_2$ means a depth-wise convolution ($d$) using a kernel size of $k_1 \times k_2$. We extract multi-scale features using convolutions and then utilize them as attention weights to reweigh the input of MSCA.

## 3   Method

In this section, we describe the architecture of the proposed SegNeXt in detail. Basically, we adopt an encoder-decoder architecture following most previous works, which is simple and easy to follow.

### 3.1   Convolutional Encoder

We adopt the pyramid structure for our encoder following most previous work [90, 4, 20]. For the building block in our encoder, we adopt a similar structure to that of ViT [16, 90] but what is different is that we do not use the self-attention mechanism but design a novel multi-scale convolutional attention (MSCA) module. As depicted in Fig. 2 (a), MSCA contains three parts: a depth-wise convolution to aggregate local information, multi-branch depth-wise strip convolutions to capture multi-scale context, and an $1 \times 1$ convolution to model relationship between different channels. The output of the $1 \times 1$ convolution is used as attention weights directly to reweigh the input of MSCA. Mathematically, our MSCA can be written as:

$$\text{Att} = \text{Conv}_{1\times1}(\sum_{i=0}^{3} \text{Scale}_i(\text{DW-Conv}(F))), \qquad (1)$$

$$\text{Out} = \text{Att} \otimes F. \qquad (2)$$

where $F$ represents the input feature. Att and Out are the attention map and output, respectively. $\otimes$ is the element-wise matrix multiplication operation. DW-Conv denotes depth-wise convolution and Scale$_i$, $i \in \{0, 1, 2, 3\}$, denotes the $i$th branch in Fig. 2(b). Scale$_0$ is the identity connection. Following [69], in each branch, we use two depth-wise strip convolutions to approximate standard depth-wise convolutions with large kernels. Here, the kernel size for each branch is set to 7, 11, and 21, respectively. The reasons why we choose depth-wise strip convolutions are two-fold. On one hand, strip convolution is lightweight. To mimic a standard 2D convolution with kernel size $7 \times 7$, we only need a pair of $7 \times 1$ and $1 \times 7$ convolutions. On the other hand, there are some strip-like objects, such as human and telephone pole in the segmentation scenes. Thus, strip convolution can be a complement of grid convolutions and helps extract strip-like features [69, 30].

Stacking a sequence of building blocks yields the proposed convolutional encoder, named MSCAN. For MSCAN, we adopt a common hierarchical structure, which contains four stages with decreasing spatial resolutions $\frac{H}{4} \times \frac{W}{4}, \frac{H}{8} \times \frac{W}{8}, \frac{H}{16} \times \frac{W}{16}$ and $\frac{H}{32} \times \frac{W}{32}$. Here, $H$ and $W$ are height and width of the input image, respectively. Each stage contains a down-sampling block and a stack of building blocks as described above. The down-sampling block has a convolution with stride 2 and kernel size $3 \times 3$, followed by a batch normalization layer [36]. Note that, in each building block of MSCAN, we use batch normalization instead of layer normalization as we found batch normalization gains more for the segmentation performance.

Table 2: Detailed settings of different sizes of the proposed SegNeXt. In this table, 'e.r.' represents the expansion ratio in the feed-forward network. '$C$' and '$L$' are the numbers of channels and building blocks, respectively. 'Decoder dimension' denotes the MLP dimension in the decoder. 'Parameters' are calculated on the ADE20K dataset [111]. Due to the different numbers of the categories in different datasets, the number of parameters may change slightly.

| stage | output size | e.r. | SegNeXt-T | SegNeXt-S | SegNeXt-B | SegNeXt-L |
|---|---|---|---|---|---|---|
| 1 | $\frac{H}{4} \times \frac{W}{4} \times C$ | 8 | $C = 32, L = 3$ | $C = 64, L = 2$ | $C = 64, L = 3$ | $C = 64, L = 3$ |
| 2 | $\frac{H}{8} \times \frac{W}{8} \times C$ | 8 | $C = 64, L = 3$ | $C = 128, L = 2$ | $C = 128, L = 3$ | $C = 128 , L = 5$ |
| 3 | $\frac{H}{16} \times \frac{W}{16} \times C$ | 4 | $C = 160, L = 5$ | $C = 320, L = 4$ | $C = 320, L = 12$ | $C = 320, L = 27$ |
| 4 | $\frac{H}{32} \times \frac{W}{32} \times C$ | 4 | $C = 256, L = 2$ | $C = 512, L = 2$ | $C = 512, L = 3$ | $C = 512, L = 3$ |
| Decoder dimension | | | 256 | 256 | 512 | 1,024 |
| Parameters (M) | | | 4.3 | 13.9 | 27.6 | 48.9 |

We desgin four encoder models with different sizes, named MSCAN-T, MSCAN-S, MSCAN-B, and MSCAN-L, respectively. The corresponding overall segmentation models are termed SegNeXt-T, SegNeXt-S, SegNeXt-B, SegNeXt-L, respectively. Detailed network settings are displayed in Tab. 2.

## 3.2 Decoder

In segmentation models [90, 108, 4], the encoders are mostly pretrained on the ImageNet dataset. To capture high-level semantics, a decoder is usually necessary, which is applied upon the encoder. In this work, we investigate three simple decoder structures, which have been shown in Fig. 3. The first one, adopted in SegFormer [90], is a purely MLP-based structure. The second one is mostly adopted CNN-based models. In this kind of structure, the output of the encoder is directly used as the input to a heavy decoder head, like ASPP [4], PSP [106], and DANet [20]. The last one is the structure adopted in our SegNeXt. We aggregate features from the last three stages and use a lightweight Hamburger [22] to further model the global context. Combined with our powerful convolutional encoder, we found that using a lightweight decoder improves performance-computation efficiency.

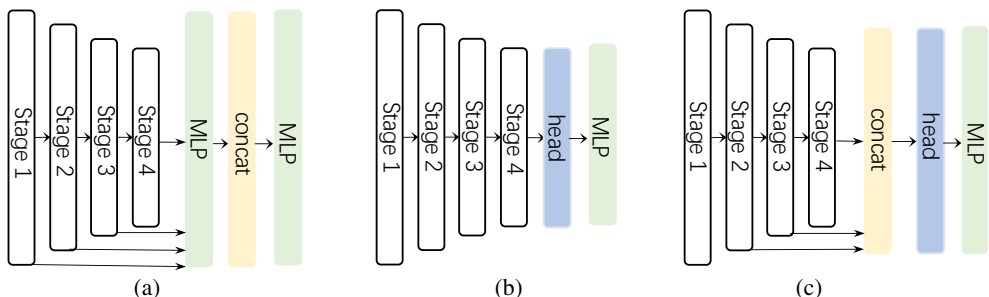

Figure 3: Three different decoder designs.

It is worth nothing that unlike SegFormer whose decoder aggregates the features from Stage 1 to Stage 4, our decoder only receives features from the last three stages. This is because our SegNeXt is based on convolutions. The features from Stage 1 contain too much low-level information and hurts the performance. Besides, operations on Stage 1 bring heavy computational overhead. In our experiment section, we will show that our convolutional SegNeXt performs much better than the recent state-of-the-art transformer-based SegFormer [90] and HRFormer [100].

## 4 Experiments

**Dataset.** We evaluate our methods on seven popular datasets, including ImageNet-1K [14], ADE20K [111], Cityscapes [12], Pascal VOC [17], Pascal Context [65], COCO-Stuff [3], and iSAID [84]. ImageNet [14] is the best-known dataset for image classification, which contains

Table 3: Comparison with state-of-the-art methods on ImageNet validation set. 'Acc.' denotes Top-1 accuracy.

| Method | Params. (M) | Acc. (%) |
|---|---|---|
| MiT-B0 [90] | 3.7 | 70.5 |
| VAN-Tiny [25] | 4.1 | 75.4 |
| **MSCAN-T** | 4.2 | **75.9** |
| MiT-B1 [90] | 14.0 | 78.7 |
| VAN-Small [25] | 13.9 | 81.1 |
| **MSCAN-S** | 14.0 | **81.2** |
| MiT-B2 [90] | 25.4 | 81.6 |
| Swin-T [58] | 28.3 | 81.3 |
| ConvNeXt-T [59] | 28.6 | 82.1 |
| VAN-Base [25] | 26.6 | 82.8 |
| **MSCAN-B** | 26.8 | **83.0** |
| MiT-B3 [28] | 45.2 | 83.1 |
| Swin-S [58] | 49.6 | 83.0 |
| ConvNeXt-S [58] | 50.1 | 83.1 |
| VAN-Large [25] | 44.8 | **83.9** |
| **MSCAN-L** | 45.2 | **83.9** |

Table 4: Comparison with state-of-the-art methods on the remote sensing dataset iSAID. Single-scale (SS) test is applied by default. Our SegNeXt-T has achieved state-of-the-art performance.

| Method | Backbone | mIoU (%) |
|---|---|---|
| DenseASPP [95] | ResNet50 | 57.3 |
| PSPNet [106] | ResNet50 | 60.3 |
| SemanticFPN [40] | ResNet50 | 62.1 |
| RefineNet [54] | ResNet50 | 60.2 |
| HRNet [79] | HRNetW-18 | 61.5 |
| GSCNN [76] | ResNet50 | 63.4 |
| SFNet [49] | ResNet50 | 64.3 |
| RANet [66] | ResNet50 | 62.1 |
| PointRend [41] | ResNet50 | 62.8 |
| FarSeg [109] | ResNet50 | 63.7 |
| UperNet [89] | Swin-T | 64.6 |
| PointFlow [47] | ResNet50 | 66.9 |
| SegNeXt-T | MSCAN-T | 68.3 |
| SegNeXt-S | MSCAN-S | 68.8 |
| SegNeXt-B | MSCAN-B | 69.9 |
| SegNeXt-L | MSCAN-L | **70.3** |

1,000 categories. Similar to most segmentation methods, we use it to pretrain our MSCAN encoder. ADE20K [111] is a challenging dataset which contains 150 semantic classes. It consists of 20,210/2,000/3,352 images in the training, validation and test sets. Cityscapes [12] mainly focuses on urban scenes and contains 5.000 high-resolution images with 19 categories. There are 2,975/500/1,525 images for training, validation and testing, respectively. Pascal VOC [17] involves 20 foreground classes and a background class. After augmentation, it has 10, 582/1, 449/1, 456 images for training, validation and testing, respectively. Pascal Context [65] contains 59 foreground classes and a background class. The training set and validation set contain 4,996 and 5,104 images, respectively. COCO-Stuff [3] is also a challenging benchmark, which contains 172 semantic categories and 164k images in total. iSAID [84] is a large-scale aerial image segmentation benchmark, which includes 15 foreground classes and a background class. Its training, validation and test sets separately involve 1,411/458/937 images.

**Implementation details.** We conduct experiments by using Jittor [32] and Pytorch [68]. Our implementation is based on timm (Apache-2.0) [85] and mmsegmentation (Apache-2.0) [11] libraries for classification and segmentation, respectively. All encoders of our segmentation models are pretrained on the ImageNet-1K dataset [14]. We adopt Top-1 accuracy and mean Intersection over Union (mIoU) as our evaluation metrics for classification and segmentation, respectively. All models are trained on a node with 8 RTX 3090 GPUs.

For ImageNet pretraining, our data augmentation method and training settings are the same as DeiT [78]. For segmentation experiments, we adopt some common data augmentation including random horizontal flipping, random scaling (from 0.5 to 2) and random cropping. The batch size is set to 8 for the Cityscapes dataset and 16 for all the other datasets. AdamW [61] is applied to train our models. We set the initial learning rate as 0.00006 and employ the poly-learning rate decay policy. We train our model 160K iterations for ADE20K, Cityscapes and iSAID datasets and 80K iterations for COCO-Stuff, Pascal VOC and Pascal Context datasets. During testing, we use both the single-scale (SS) and multi-scale (MS) flip test strategies for a fair comparison. More details can be found in our supplementary materials.

## 4.1 Encoder Performance on ImageNet

ImageNet pretraining is a common strategy for training segmentation models [106, 5, 90, 100, 4]. Here, we compare the performance of our MSCAN with several recent popular CNN-based and transformer-based classification models. As shown in Tab. 3, our MSCAN achieves better results than the recent state-of-the-art CNN-based method, ConvNeXt [59] and outperforms popular transformer-based methods, like Swin Transformer [58] and MiT, the encoder of SegFormer [90].

Table 5: Performance of different attention mechanisms in decoder. SegNeXt-B w/ Ham means the MSCAN-B encoder plus the Ham decoder. FLOPs are calculated using the input size of 512×512.

| Architecture | Params. (M) | GFLOPs | mIoU (SS) | mIoU (MS) |
|---|---|---|---|---|
| SegNeXt-B w/ CC [35] | 27.8 | 35.7 | 47.3 | 48.6 |
| SegNeXt-B w/ EMA [46] | 27.4 | 32.3 | 48.0 | 49.1 |
| SegNeXt-B w/ NL [83] | 27.6 | 40.9 | 48.6 | 50.0 |
| SegNeXt-B w/ Ham [22] | 27.6 | 34.9 | 48.5 | 49.9 |

## 4.2 Ablation study

**Ablation on MSCA design.** We conduct ablation study on MSCA design on both ImageNet and ADE20K dataset. K × K branch contains a depth-wise 1 × K convolution and a K × 1 depth-wise convolution. 1 × 1 conv means the channel mixing operation. Attention means the element-wise product, which makes the network obtain adaptive ability. Results are shown in Tab. 6. We can find that each part contributes to the final performance.

Table 6: Ablation study on the design of MSCA. Top-1 means Top-1 accuracy on ImageNet dataset and mIoU denotes mIoU on ADE20K benchmark. The results are based on MSCAN-T.

| 7 × 7 branch | 11 × 11 branch | 21 × 21 branch | 1 × 1 Conv | Attention | Top-1 | mIoU |
|---|---|---|---|---|---|---|
| ✓ | ✗ | ✗ | ✓ | ✓ | 74.7 | 39.6 |
| ✗ | ✓ | ✗ | ✓ | ✓ | 75.2 | 39.7 |
| ✗ | ✗ | ✓ | ✓ | ✓ | 75.3 | 40.0 |
| ✓ | ✓ | ✓ | ✗ | ✓ | 74.8 | 39.1 |
| ✓ | ✓ | ✓ | ✓ | ✗ | 75.5 | 40.5 |
| ✓ | ✓ | ✓ | ✓ | ✓ | **75.9** | **41.1** |

**Global Context for Decoder.** Decoder plays an important role in integrating global context from multi-scale features for segmentation models. Here, we investigate the influence of different global context modules on decoder. As shown in most previous works [83, 20], attention-based decoders achieves better performance for CNNs than pyramid structures [106, 4], we thus only show the results using attention-based decoders. Specifically, we show results with 4 different types of attention-based decoders, including non-local (NL) attention [83] with $\mathcal{O}(n^2)$ complexity and CCNet [35], EMANet [46], and HamNet [22] with $\mathcal{O}(n)$ complexity. As shown in Tab. 5, Ham achieves the best trade-off between complexity and performance. Therefore, we use Hamburger [22] in our decoder.

Table 7: Performance of different decoder structures. SegNeXt-T (a) means Fig. 3 (a) is used in decoder. FLOPs are calculated using the input size of 512×512. SegNeXt-T (c) w/ stage 1 means the output of stage 1 is also sent into the decoder.

| Architecture | Params. (M) | GFLOPs | mIoU (SS) | mIoU (MS) |
|---|---|---|---|---|
| SegNeXt-T (a) | 4.4 | 10.0 | 40.3 | 41.1 |
| SegNeXt-T (b) | 4.2 | 4.9 | 30.9 | 40.6 |
| SegNeXt-T (c) | 4.3 | 6.6 | 41.1 | 42.2 |
| SegNeXt-T (c) w/ stage 1 | 4.3 | 12.1 | 40.7 | 42.2 |

**Decoder Structure.** Unlike image classification, segmentation models need high-resolution outputs. We ablate three different decoder designs for segmentation, all of which have been shown in Fig. 3. The corresponding results are listed in Tab. 7. We can see that SegNeXt (c) achieves the best performance and the computational cost is also low.

**Importance of Our MSCA.** Here, we conduct experiments to demonstrate the importance of MSCA for segmentation. As a comparison, we follow VAN [25] and replace the multiple branches in our MSCA with a single convolution with a large kernel. As shown in Tab. 8 and Tab. 3, we can observe that though the performance of the two encoders is close in ImageNet classification, SegNeXt

Table 8: Importance of our multi-scale convolutional attention (MSCA). SegNeXt-T w/o MSCA means we use only a branch with a large kernel convolution as done in [25] to replace the multiple branches in our MSCA. FLOPs are calculated using the input size of $512 \times 512$.

| Architecture | Params. (M) | GFLOPs | mIoU (SS) | mIoU (MS) |
|---|---|---|---|---|
| SegNeXt-T w/o MSCA | 4.2 | 6.5 | 39.5 | 40.9 |
| SegNeXt-T w/ MSCA | 4.3 | 6.6 | 41.0 | 42.5 |
| SegNeXt-S w/o MSCA | 13.8 | 15.8 | 43.5 | 45.2 |
| SegNeXt-S w/ MSCA | 13.9 | 15.9 | 44.3 | 45.8 |

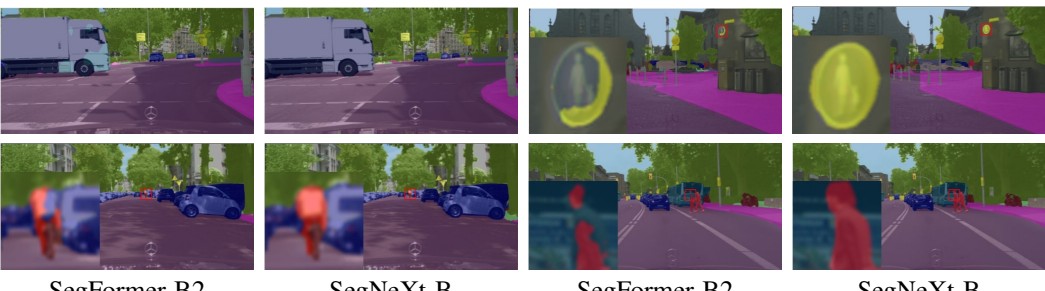

| SegFormer-B2 | SegNeXt-B | SegFormer-B2 | SegNeXt-B |
|---|---|---|---|

Figure 4: Qualitative Comparison of SegNeXt-B and SegFormer-B2 on the Cityscapes dataset. More visual results can be found in our supplementary materials.

w/ MSCA yields much better results than the setting w/o MSCA. This indicates that aggregating multi-scale features is crucial in encoder for semantic segmentation.

## 4.3 Comparison with state-of-the-art methods

In this subsection, we compare our method with state-of-the-art CNN-based methods, such as HRNet [79], ResNeSt [104], and EfficientNet [77], and transformer-based methods, like Swin Transformer [58], SegFormer [90], HRFormer [100], MaskFormer [10], and Mask2Former [9].

**Performance-computation trade-off.** ADE20K and Cityscapes are two widely used benchmarks in semantic segmentation. As shown in Fig. 1, we plot the performance-computation curves of different methods on the Cityscape and ADE20K validation set. Clearly, our method achieves the best trade-off between performance and computations compared to other state-of-the-art methods, like SegFormer [90], HRFormer [100], and MaskFormer [10].

**Comparison with state-of-the-art transformers.** We compare SegNeXt with state-of-the-art transformer models on the ADE20K, Cityscapes, COCO-Stuff and Pascal Context benchmarks. As shown in Tab. 9, SegNeXt-L surpasses Mask2Former with Swin-T backbone by 3.3 mIoU (51.0 v.s. 47.7) with similar parameters and computational cost on he ADE20K dataset. Moreover, SegNeXt-B yields 2.0 mIoU improvement (48.5 v.s. 46.5) compared to SegFormer-B2 using only 56% computations on the ADE20K dataset. In particular, since the self-attention in SegFormer [90] is of quadratic complexity *w.r.t.,* the input size while our method uses convolutions, this makes our method perform greatly well when dealing with high-resolution images from the Cityscapes dataset. For instance, SegNeXt-B gains 1.6 mIoU (81.0 v.s. 82.6) over SegFormer-B2 but uses 40% less computations. In Fig. 4, we also show a qualitative comparison with SegFormer. We can see that thanks to the proposed MSCA, our method recognizes well when processing object details.

**Comparison with state-of-the-art CNNs.** As shown in Tab. 4, Tab. 10, and Tab. 12, we compare our SegNeXt with state-of-the-art CNNs such as ResNeSt-269 [104], EfficientNet-L2 [112], and HRNet-W48 [79] on the Pascal VOC 2012, Pascal Context, and iSAID datasets. SegNeXt-L outperforms the popular HRNet (OCR) [79, 99] model (60.3 v.s. 56.3) using even less parameters and computations, which is elaborately designed for the segmentation task. Moreover, SegNeXt-L performs even better than EfficientNet-L2 (NAS-FPN), which is pretrained on additional 300 million unavailable images, on the Pascal VOC 2012 test leaderboard. It is worth noting that EfficientNet-L2 (NAS-FPN) has 485M parameters, while SegNeXt-L has only 48.7M parameters.

Table 9: Comparison with state-of-the-art methods on the ADE20K, Cityscapes and COCO-Stuff benchmarks. The number of FLOPs (G) is calculated on the input size of 512×512 for ADE20K and COCO-Stuff, and 2,048×1,024 for Cityscapes. † means models pretrained on ImageNet-22K.

| Model | Params (M) | ADE20K GFLOPs | mIoU (SS) | (MS) | Cityscapes GFLOPs | mIoU (SS) | (MS) | COCO-Stuff GFLOPs | mIoU (SS) | (MS) |
|---|---|---|---|---|---|---|---|---|---|---|
| Segformer-B0 [90] | 3.8 | 8.4 | 37.4 | 38.0 | 125.5 | 76.2 | 78.1 | 8.4 | 35.6 | - |
| SegNeXt-T | 4.3 | 6.6 | **41.1** | **42.2** | 50.5 | **79.8** | **81.4** | 6.6 | **38.7** | **39.1** |
| Segformer-B1 [90] | 13.7 | 15.9 | 42.2 | 43.1 | 243.7 | 78.5 | 80.0 | 15.9 | 40.2 | - |
| HRFormer-S [100] | 13.5 | 109.5 | 44.0 | 45.1 | 835.7 | 80.0 | 81.0 | 109.5 | 37.9 | 38.9 |
| SegNeXt-S | 13.9 | 15.9 | **44.3** | **45.8** | 124.6 | **81.3** | **82.7** | 15.9 | **42.2** | **42.8** |
| Segformer-B2 [90] | 27.5 | 62.4 | 46.5 | 47.5 | 717.1 | 81.0 | 82.2 | 62.4 | 44.6 | - |
| MaskFormer [10] | 42 | 55 | 46.7 | 48.8 | - | - | - | - | - | - |
| SegNeXt-B | 27.6 | 34.9 | **48.5** | **49.9** | 275.7 | **82.6** | **83.8** | 34.9 | **45.8** | **46.3** |
| SETR-MLA†[108] | 310.6 | - | 48.6 | 50.1 | - | 79.3 | 82.2 | - | - | - |
| DPT-Hybrid [70] | 124.0 | 307.9 | - | 49.0 | - | - | - | - | - | - |
| Segformer-B3 [90] | 47.3 | 79.0 | 49.4 | 50.0 | 962.9 | 81.7 | 83.3 | 79.0 | 45.5 | - |
| Mask2Former [9] | 47 | 74 | 47.7 | 49.6 | - | - | - | - | - | - |
| HRFormer-B [100] | 56.2 | 280.0 | 48.7 | 50.0 | 2223.8 | 81.9 | 82.6 | 280.0 | 42.4 | 43.3 |
| MaskFormer [10] | 63 | 79 | 49.8 | 51.0 | - | - | - | - | - | - |
| SegNeXt-L | 48.9 | 70.0 | **51.0** | **52.1** | 577.5 | **83.2** | **83.9** | 70.0 | **46.5** | **47.2** |

Table 10: Comparison with state-of-the-art methods on Pascal VOC dataset. * means COCO [55] pretraining. † denotes JFT-300M [74] pretraining. $ utilizes additional 300M unlabeled images for pretraining.

| Method | Backbone | mIoU |
|---|---|---|
| DANet [20] | ResNet101 | 82.6 |
| OCRNet [99] | HRNetV2-W48 | 84.5 |
| HamNet [22] | ResNet101 | 85.9 |
| EncNet* [103] | ResNet101 | 85.9 |
| EMANet* [46] | ResNet101 | 87.7 |
| DeepLabV3+* [7] | Xception-71 | 87.8 |
| DeepLabV3+† [7] | Xception-JFT | 89.0 |
| NAS-FPN$ [112] | EfficientNet-L2 | 90.5 |
| SegNeXt-T | MSCAN-T | 82.7 |
| SegNeXt-S | MSCAN-S | 85.3 |
| SegNeXt-B | MSCAN-B | 87.5 |
| SegNeXt-L* | MSCAN-L | **90.6** |

Table 11: Comparison with state-of-the-art real-time methods on Cityscapes test dataset. We test our method with a single RTX-3090 GPU and AMD EPYC 7543 32-core processor CPU . Without using any optimizations, SegNeXt-T can achieve 25 frames per second (FPS), which meets the requirements of real-time applications.

| Method | Input size | mIoU |
|---|---|---|
| ESPNet [62] | 512×1,024 | 60.3 |
| ESPNetv2 [63] | 512×1,024 | 66.2 |
| ICNet [105] | 1,024 × 2,048 | 69.5 |
| DFANet [45] | 1,024 × 1,024 | 71.3 |
| BiSeNet [97] | 768 × 1,536 | 74.6 |
| BiSeNetv2 [96] | 512 × 1,024 | 75.3 |
| DF2-Seg [52] | 1,024 × 2,048 | 74.8 |
| SwiftNet [67] | 1,024 × 2,048 | 75.5 |
| SFNet [49] | 1,024 × 2,048 | 77.8 |
| SegNeXt-T | 768 × 1,536 | **78.0** |

**Comparison with real-time methods.** In addition to the state-of-the-art performance, our method is also suitable for real-time deployments. Even without any specific software or hardware acceleration, SegNeXt-T realizes 25 frames per second (FPS) using a single 3090 RTX GPU when dealing with an image of size 768×1,536. As shown in Tab. 11, our method sets new state-of-the-art results for real-time segmentation on the Cityscapes test set.

## 4.4 Weakly-Supervised Semantic Segmentation

In this subsection, we apply the proposed network to the weakly-supervised semantic segmentation task. In this task, a pseudo segmentation map is often generated by a classification model using CAM [110]. Previous works mostly utilize VGGNet [72] or ResNets [28, 87] as the CAM generator. Here, we test the performance of the CAMs produced by our MSCAN. We use the EPS [43] architecture and follow the training strategies and recipes. The numerical results are shown in Tab. 13. We can see that simply replacing the ResNet38 backbone with our MSCAN can clearly improve the performance compared to the EPS baseline. When using our SegNeXt as the segmentation network, the performance gain increases further.

Table 12: Comparison on Pascal Context benchmark. The number of FLOPs is calculated with the input size of 512×512. * means ImageNet-22K pretraining. † denotes ADE20K pretraining.

| Method | Backbone | Params.(M) | GFLOPs | mIoU (SS/MS) | |
|---|---|---|---|---|---|
| DANet [20] | ResNet101 | 69.1 | 277.7 | - | 52.6 |
| EMANet [46] | ResNet101 | 61.1 | 246.1 | - | 53.1 |
| HamNet [22] | ResNet101 | 69.1 | 277.9 | - | 55.2 |
| HRNet (OCR) [79] | HRNetW48 | 74.5 | - | - | 56.2 |
| DeepLabV3+ [7] | ResNeSt-269 | - | - | - | 58.9 |
| SETR-MLA* [108] | ViT-Large | 309.5 | - | 54.9 | 55.8 |
| HRFormer-B [100] | HRFormer-B | 56.2 | 280.0 | 57.6 | 58.5 |
| DPT-Hybrid† [70] | ViT-Hybrid | 124.0 | - | - | 60.5 |
| SegNeXt-T | MSCAN-T | 4.2 | 6.6 | 51.2 | 53.3 |
| SegNeXt-S | MSCAN-S | 13.9 | 15.9 | 54.2 | 56.1 |
| SegNeXt-B | MSCAN-B | 27.6 | 34.9 | 57.0 | 59.0 |
| SegNeXt-L | MSCAN-L | 48.8 | 70.0 | 58.7 | 60.3 |
| SegNeXt-L† | MSCAN-L | 48.8 | 70.0 | **59.2** | **60.9** |

Table 13: Comparisons to previous state-of-the-art weakly-supervised semantic segmentation approaches on the PASCAL VOC 2012 validation set. All the segmentation results are based on the ResNet backbone [28, 87] except ours which utilize MSCAN-B.

| Methods | Network | Supervision | mIoU (%) on Val. |
|---|---|---|---|
| FickleNet$_{2019}$ [42] | DeeplabV2 | Image + Saliency | 64.9 |
| OAA$^+_{2019}$ [37] | DeeplabV1 | Image + Saliency | 65.2 |
| ICD$_{2020}$ [18] | DeeplabV1 | Image + Saliency | 67.8 |
| Multi-Est.$_{2020}$ [19] | DeeplabV1 | Image + Saliency | 67.2 |
| DRS$_{2021}$ [39] | DeeplabV2 | Image + Saliency | 71.2 |
| Group-WSSS$_{2021}$ [53] | DeeplabV2 | Image + Saliency | 68.2 |
| AuxSegNet$_{2021}$ [92] | DeeplabV1 | Image + Saliency | 69.0 |
| EDAM$_{2021}$ [86] | DeeplabV1 | Image + Saliency | 70.9 |
| EPS$_{2021}$ [43] | DeeplabV2 | Image + Saliency | 70.9 |
| L2G$_{2022}$ [38] | DeeplabV1 | Image + Saliency | 72.0 |
| MSCAN + EPS [43] (Ours) | DeeplabV2 | Image + Saliency | 71.7 |
| MSCAN + EPS [43] (Ours) | SegNeXt | Image + Saliency | **72.2** |

## 5 Conclusions and Discussion

In this paper, we analyze previous successful segmentation models and find the good characteristics owned by them. Based on the findings, we present a tailored convolutional attention module MSCA and a CNN-style network SegNeXt. Experimental results demonstrate that SegNeXt surpasses current state-of-the-art transformer-based methods by a considerable margin.

Recently, transformer-based models have dominated various segmentation leaderboards. Instead, this paper shows that CNN-based methods can still perform better than transformer-based methods when using a proper design. We hope this paper could encourage researchers to further investigate the potential of CNNs.

Our model also has its limitations, for example, extending this method to large-scale models with 100M+ parameters and the performance on other vision or NLP tasks. These will be addressed in our future works.

## Acknowledgment

This work was supported by the National Key R&D Program of China (NO. 2018AAA0100400) and the Natural Science Foundation of China (No. 62220106003, No. 62176130, and No. 62276145). We would like to thank Yi Zhang and Zhengyang Geng for their kind help in experiments.

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
