# Appendix for "SegNeXt: Rethinking Convolutional Attention Design for Semantic Segmentation"

**Meng-Hao Guo**[1]    **Cheng-Ze Lu**[2]    **Qibin Hou**[2]    **Zheng-Ning Liu**[3]
**Ming-Ming Cheng**[2]    **Shi-Min Hu**[1]

[1]BNRist, Department of Computer Science and Technology, Tsinghua University
[2]TMCC, CS, Nankai University        [3]Fitten Tech, Beijing, China

## 1   Training Details

We show some training details on different datasets omitted in the main paper in Tab. 1. For different benchmarks, we employ different training settings for fair comparison.

Table 1: Training details on different benckmarks. 80K + 80K means we pretrain 80K iterations on Pascal VOC trainaug set and finetune 80K on its trainval set. 80K + 40K denotes we pretrain 600K iterations on COCO dataset and finetune 40K on its trainval set.

| Dataset | Crop Size | Batch Size | Iterations |
|---|---|---|---|
| ADE20K [10] | $512 \times 512$ | 16 | 160K |
| Cityscapes [2] | $1,024 \times 1,024$ | 8 | 160K |
| COCO-Stuff [1] | $512 \times 512$ | 16 | 80K |
| Pascal VOC [3] | $512 \times 512$ | 16 | 80K + 80K |
| Pascal VOC [3] w/ COCO [5] | $512 \times 512$ | 16 | 600K + 40K |
| Pascal Context  [7] | $480 \times 480$ | 16 | 80K |
| iSAID [9] | $896 \times 896$ | 16 | 160K |

## 2   Ablation about MSCA Head

In addition to using a variant of self-attention as our head, we also used MSCA as our head. Results in Tab. 2 show Ham head [4] achieves a better performance than MSCA head, which demonstrates a CNN-style encoder requires a segmentation head with a global receptive field.

## 3   More Qualitative Results

In the main paper, we show the qualitative results on Cityscapes dataset. Here, we display qualitative results on ADE20K dataset in  Fig. 1. The figure clearly shows that our method is better at understanding the details.

## 4   Visualization results

We adopt Grad-CAM [8] to conduct visualization. As shown in  Fig. 2, we can clearly find our MSCAN shows better visualization results. In particular, when object occupies most of area in an image (shown in first three columns) or multiple objects in an image (shown in last three columns), ConvNeXt [6] appears inaccurate, while our MSCAN still works well. It shows the effectiveness of larger receptive field and multi-scale information aggregation.

36th Conference on Neural Information Processing Systems (NeurIPS 2022).

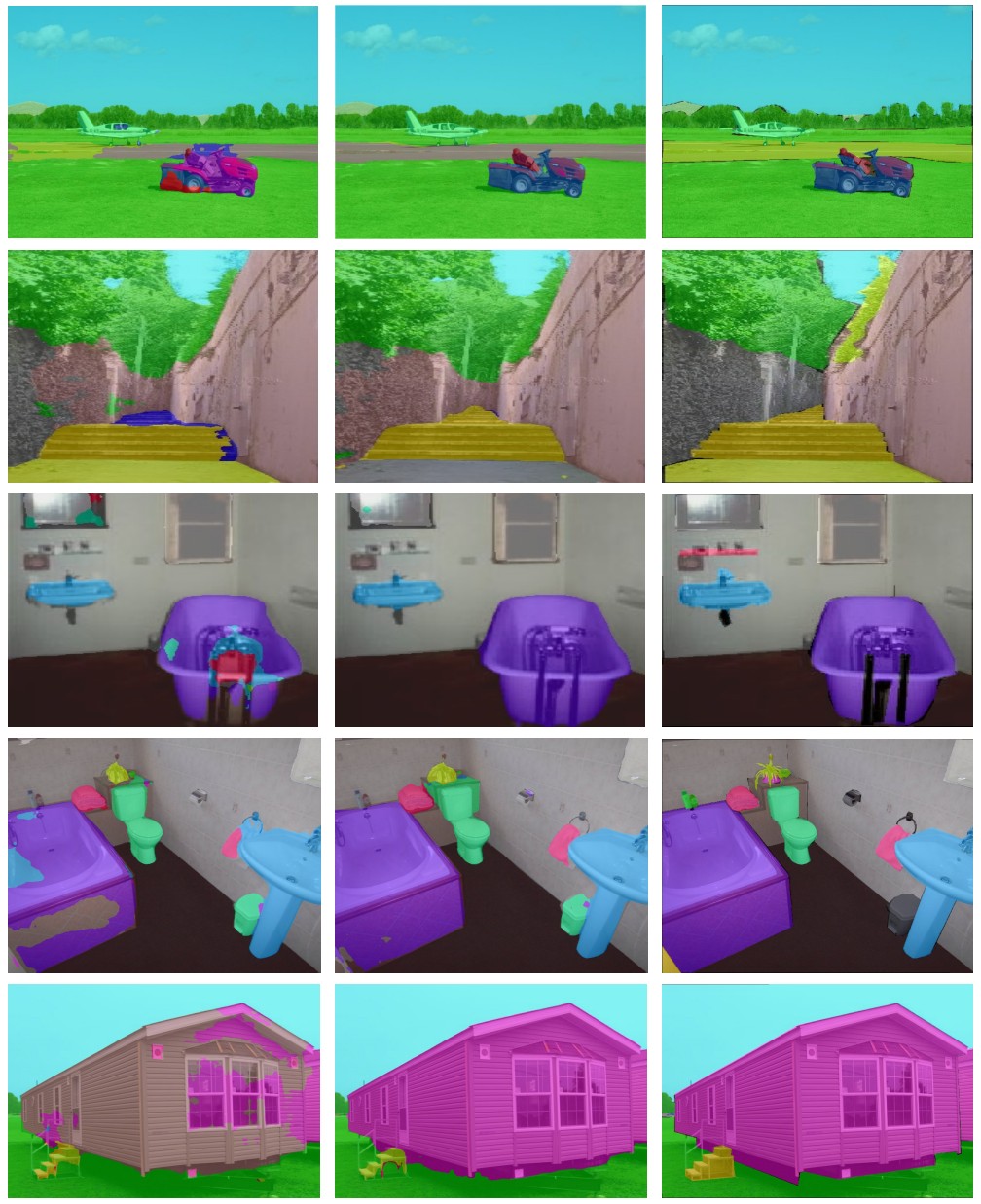

Figure 1: Qualitative results on ADE20K dataset. Left: SegFromer-B2. Middle: SegNeXt-B. Right: Ground trurh.

Table 2: Performance of different head in decoder. SegNeXt-T w/ Ham means the MSCAN-T encoder plus the Ham decoder. FLOPs are calculated using the input size of 512×512. Experiments are conducted on COCO-Stuff dataset.

| Architecture | Params. (M) | GFLOPs | mIoU (SS) | mIoU (MS) |
|---|---|---|---|---|
| SegNeXt-T w/ MSCA | 4.4 | 6.7 | 38.2 | 38.6 |
| SegNeXt-T w/ Ham [4] | 4.3 | 6.6 | 38.7 | 39.1 |
| SegNeXt-S w/ MSCA | 14.0 | 15.9 | 42.1 | 42.4 |
| SegNeXt-S w/ Ham [4] | 13.9 | 15.9 | 42.2 | 42.8 |
| SegNeXt-B w/ MSCA | 28.0 | 33.6 | 45.1 | 45.5 |
| SegNeXt-B w/ Ham [4] | 27.6 | 34.9 | 45.8 | 46.3 |
| SegNeXt-L w/ MSCA | 50.1 | 69.8 | 45.9 | 46.4 |
| SegNeXt-L w/ Ham [4] | 48.9 | 70.0 | 46.5 | 47.2 |

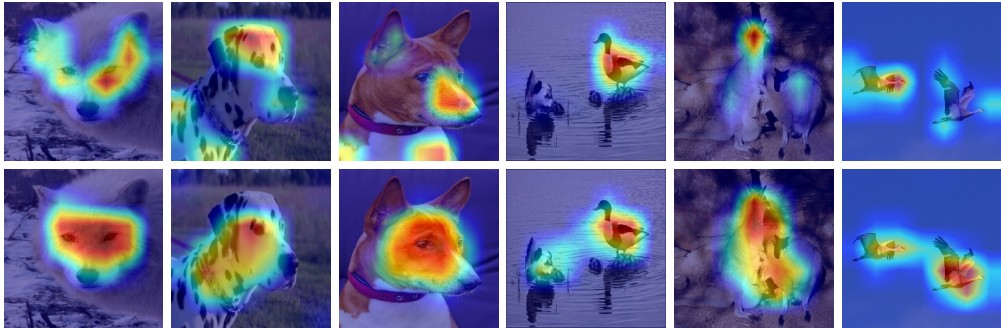

Figure 2: Visualization results by using Grad-CAM [8]. Top: grad-cam figures of ConvNeXt [6]. Bottom: grad-cam figures of our MSCAN.