# OpenReview forum: "SegNeXt: Rethinking Convolutional Attention Design for Semantic Segmentation"
_NeurIPS.cc/2022/Conference — NeurIPS 2022 Accept_

### Official Review · Reviewer_S4w5 · 2022-06-24

**Rating:** 6
**Confidence:** 5
**Soundness:** 2 fair
**Presentation:** 3 good
**Contribution:** 3 good

**Summary:**


This paper proposes a simple convolution network SegNeXt for semantic segmentation which include three important properties : strong encoder, multiscale interaction and spatial attention. It shows convolutional attention is a more efficient and effective way to encode contextual information than the self-attention mechanism in transformers. SegNeXt improves the performance of previous state-of-the-art methods on popular benchmarks, including ADE20K, Cityscapes, COCO-Stuff, Pascal VOC, Pascal Context, and iSAID.


**Questions:**

See the weakness part.

**Ethics Review Area:**

["I don’t know"]

**Strengths And Weaknesses:**


Strength:
1, The paper is well written and easy to follow for both method and experiment part. The summary of Table.1 is interesting.

2. The proposed approach achieve new state-of-the-art results on iSAID, ADE-20k, COCO-stuff and Pascal Context dataset compared with recent transformer based approach. The result in Figure.1 looks good.

3, The proposed approach: convolutional attention is simple yet effective. MSCA is an improved version of large kernel design for segmentation.

4, The results on Pascal VOC dataset rank the first.

Weakness:

1, The technical novelty is very limited. For example, despite the MSCA is simple, it follows the large kernel design which has been proposed in previous works. [1] [2]

Moreover. The decoder adopts multi-scale feature fusion via Hamburger which is a previous work.

Thus the technical novelty makes the submission looks like a report.

2, Could not find new insights. The ablation studies are not good which lacks design analysis of component. A better comparison with HRFormer or HRnet is needed to show the effectiveness of spatial attention.


3, In Tab10. Why not list the FPS of other method for better comparison.


4, Missing ablation on MSCA design on both ImageNet or other segmentation datasets.



[1]  Large kernel matters–improve semantic segmentation by global convolutional network. CVPR-2017.

[2] Convnext -CVPR-2022

[3] Is attention better than matrix decomposition? ICLR-2021.

---

> ### Author Response · Authors · 2022-08-02
> **Response to Reviewer S4w5(1/2)**
>
> Dear Reviewer S4w5：
>
> Thanks for your effort for reviewing our paper and giving some kind suggestions. We are happy to see your positive comments on writting, experiments and method. We hope the following responses could solve your concerns.
>
> * For Question1: "The technical novelty is very limited."
>
>   This paper is not a simple combination of ConvNeXt[1] and segmentation task. We do not deny we are inspired from ConvNeXt[1], large kernel matters[2] and HamNet[3], which are included in our reference list. Here, we illustrate the difference between SegNeXt and others from three aspects: analysis, visualization and experimental results.
>
>   * Analysis:  The goal of this paper is to find the simple and effective operation / network for segmentation task. To achieve the target, we analyze and summarize the expected properties of segmentation task such as multi-scale information, strong encoder, attention mechanism (dynamic process) and low complexity. For common CNNs such as ConvNeXt[1] and Large Kernel matters[2], they only satisfy the strong encoder and low complexity, but ignores the multi-scale information and attention mechanism, which are critical for segmentation task.  For common transformers such as SegFormer[4] and SETR[5], they also ignore multi-scale information and low complexity especially when dealing with high resolution images.  As for our MSCAN, it satisfies all listed properties for segmentation, which is a simple, suitable and new network for semantic segmentation.  Besides, we also present visualizations and numerical experiments to support our claims in the following response. Besides, as we known, CNN-based encoder is lacking in global receptive field and we hope our architecture can avoid it in the decoder stage. So, we choose one of global module Ham[3] to solve this problem. A simple and suitable encoder for segmentation task is what we claim for our main contribution.
>   * Visualization: We adopt Grad-CAM[6] to conduct visualization to prove the effectiveness. Due to the limitation of openreview, we add visualization results in the supplementary material. From the visualization results, we can easily find two shortcomings of convnext: Insufficient receptive field and lack of multi-scale information.
>   * Experimental results: We replace our backbone with ConvNeXt and fairly compare ConvNeXt with our MSCAN on ADE20K dataset. As shown in following table, experiments also indicate the superiority of our method.
>
>   | Method              | Params.(M) | mIoU(SS) | mIoU(MS) |
>   | ------------------- | ---------- | -------- | -------- |
>   | SegNeXt w/ ConvNeXt | 28.4       | 43.2     | 44.5     |
>   | SegNeXt w/ MSCAN    | 27.6       | 48.5     | 49.9     |
>   | SegNeXt w/ ConvNeXt | 50.0       | 46.2     | 47.6     |
>   | SegNeXt w/ MSCAN    | 48.9       | 51.0     | 52.1     |
>
> * For Question2 "A better comparison with HRFormer or HRnet is needed to show the effectiveness of spatial attention."
>
>   We conduct experiments to compare HRNet and HRFormer with our MSCAN on ADE20K dataset. Results are shown in the following table. Our method significantly surpasses HRNet and HRFormer with similar size, which demonstrate the superiority of our method.
>
>   | Method           | Params.(M) | mIoU(SS) | mIoU(MS) |
>   | ---------------- | ---------- | -------- | -------- |
>   | SegNeXt w/ HRNet | 9.9        | 38.1     | 39.4     |
>   | SegNeXt w/ MSCAN | 4.3        | 41.1     | 42.2     |
>   | HRFormer-B       | 56.2       | 48.7     | 50.0     |
>   | SegNeXt w/ HRNet | 65.7       | 43.0     | 44.6     |
>   | SegNeXt w/ MSCAN | 48.9       | 51.0     | 52.1     |
>
>
> To be continued.

---

> > ### Author Response · Authors · 2022-08-02
> > **Response to Reviewer S4w5(2/2)**
> >
> > * For Question3 "In Tab10. Why not list the FPS of other method for better comparison."
> >
> >   There are three main reasons.
> >
> >   * The first reason is the difference in hardware. Different hardware will produce different throughputs.
> >   * The second one is that different methods adopt different software optimization such as SFNet[7] using TensorRT to accelerate algorithm.  The above reasons cause an unfair comparison if we listed FPS.
> >   * The last reason is that we consider 25FPS a threshold for real-time applications. Thus, we compared methods that can achieve above 25 FPS.
> >
> > * For Question4 "Missing ablation on MSCA design on both ImageNet or other segmentation datasets."
> >
> >   We add ablation study on MSCA design on both ImageNet and ADE20K dataset. K x K branch contains a depth-wise 1xk convolution and a kx1 depth-wise convolution. 1x1 conv means the channel mixing operation. Attention means the element-wise product, which makes the network obtain the adaptive ability.
> >
> >   | 7x7 branch | 11x11 branch | 21x21 branch | 1x1 conv | Attention | Top-1 Acc. (%) | mIoU |
> >   | :--------: | :----------: | :----------: | :------: | :-------: | :------------: | :--: |
> >   |     ✔      |    **X**     |    **X**     |    ✔     |     ✔     |      74.7      | 39.6 |
> >   |   **X**    |      ✔       |    **X**     |    ✔     |     ✔     |      75.2      | 39.7 |
> >   |   **X**    |    **X**     |      ✔       |    ✔     |     ✔     |      75.3      | 40.0 |
> >   |     ✔      |      ✔       |      ✔       |  **X**   |     ✔     |      74.8      | 39.1 |
> >   |     ✔      |      ✔       |      ✔       |    ✔     |   **X**   |      75.5      | 40.5 |
> >   |     ✔      |      ✔       |      ✔       |    ✔     |     ✔     |      75.9      | 41.1 |
> >
> > ​
> >
> > [1]: Liu, Z., Mao, H., Wu, C. Y., Feichtenhofer, C., Darrell, T., & Xie, S. (2022). A convnet for the 2020s. In *Proceedings of the IEEE/CVF Conference on Computer Vision and Pattern Recognition* (pp. 11976-11986).
> >
> > [2]: Peng, C., Zhang, X., Yu, G., Luo, G., & Sun, J. (2017). Large kernel matters--improve semantic segmentation by global convolutional network. In *Proceedings of the IEEE conference on computer vision and pattern recognition* (pp. 4353-4361).
> >
> > [3]: Geng, Z., Guo, M. H., Chen, H., Li, X., Wei, K., & Lin, Z. (2021). Is attention better than matrix decomposition?. *arXiv preprint arXiv:2109.04553*.
> >
> > [4]: Xie, E., Wang, W., Yu, Z., Anandkumar, A., Alvarez, J. M., & Luo, P. (2021). SegFormer: Simple and efficient design for semantic segmentation with transformers. *Advances in Neural Information Processing Systems*, *34*, 12077-12090.
> >
> > [5]:  Zheng, S., Lu, J., Zhao, H., Zhu, X., Luo, Z., Wang, Y., ... & Zhang, L. (2021). Rethinking semantic segmentation from a sequence-to-sequence perspective with transformers. In *Proceedings of the IEEE/CVF conference on computer vision and pattern recognition* (pp. 6881-6890).
> >
> > [6]: Selvaraju, R. R., Cogswell, M., Das, A., Vedantam, R., Parikh, D., & Batra, D. (2017). Grad-cam: Visual explanations from deep networks via gradient-based localization. In *Proceedings of the IEEE international conference on computer vision* (pp. 618-626).
> >
> > [7]: Li, X., You, A., Zhu, Z., Zhao, H., Yang, M., Yang, K., ... & Tong, Y. (2020, August). Semantic flow for fast and accurate scene parsing. In *European Conference on Computer Vision* (pp. 775-793). Springer, Cham.

---

> > > ### Comment · Reviewer_S4w5 · 2022-08-05
> > > **The rebuttal solves most of my concerns**
> > >
> > > The rebuttal solves most of my concerns. I raise my rating as weak accept.
> > >
> > > Despite the novelty of paper is a little weak, I appreciate the benchmarking effort of this work for various segmentation datasets.
> > >
> > > Also, make sure to put these results(Q2. Q4) on the main paper rather than reporting benchmark results.
> > >
> > > Moreover, several paper on multi-scale fusion should also be cited.
> > >
> > > [1] Context contrasted feature and gated multi-scale aggregation for scene segmentation CVPR-2018
> > >
> > > [2]  Gated Fully Fusion for Semantic Segmentation, AAAI-2020
> > >
> > > [3] Hierarchical Multi-Scale Attention for Semantic Segmentation Arxiv-2020

---

> > > > ### Author Response · Authors · 2022-08-05
> > > > **Thanks for raising rating.**
> > > >
> > > > Thanks for appreciating our paper and rising your rating.
> > > > We will add above resuls (Q2, Q4) and cite the related papers in the main paper.

---

### Official Review · Reviewer_r2Bq · 2022-07-09

**Rating:** 5
**Confidence:** 4
**Soundness:** 2 fair
**Presentation:** 3 good
**Contribution:** 2 fair

**Summary:**

This paper presents SegNeXt, a simple convolutional network architecture for semantic segmentation. The main contribution of this paper is replacing the standard convolutions and self-attention with the spatial attention proposed in this paper. Experimental results demonstrate that SegNeXt surpasses current state-of-the-art transformer-based methods by a considerable margin.

**Questions:**

Please see the weaknesses.

**Ethics Review Area:**

["Discrimination / Bias / Fairness Concerns", "I don’t know"]

**Limitations:**

This work has described some limitations and potential negative social impact.

**Strengths And Weaknesses:**

**Strengths**:

1) This paper is well written, and I can easily understand the overall flowchart of this paper.

2) The experimental results of this paper are good.

**Weaknesses**:

Although the idea of this work seems interesting, the good performance is not sufficiently documented.

Firstly, the motivation of this paper is not clearly explained. In Line.112 to Line.137, the authors directly tell us that the pipeline proposed MSCA, while they do not tell us why to design such architecture.  This makes me confused that would multi-scale convolutional attention be really better than the self-attention mechanism?

Secondly, the authors do not provide any visualization results to show the shows the superiority of MSCA compared with self-attention.

In conclusion, I think the main drawback of this paper is the authors do not tell us that what defects of self-attention are solved by MSCA?

---

> ### Author Response · Authors · 2022-08-02
> **Response to Reviewer r2Bq**
>
> Dear Reviewer *r2Bq*：
>
> Thanks for your effort for reviewing our paper and giving some kind suggestions. We are happy to see your positive comments on writting and experiments. We hope the following responses could solve your concerns.
>
> We will try to solve your concerns from four aspects: motivation, advantages,  visualization and experimental results.
>
> * Motivation: The goal of this paper is to find the simple and effective operation / network for segmentation task. The original idea comes from recent convolution neural network ConvNeXt[1].  How to design a successful CNN-sytle network for segmentation task ? We find three shortcomings of ConvNeXt: (1) Insufficient receptive field, especially for processing high-resolution segmentation images; (2) without multi-scale information; (3) no adaptability. To achieve above properties, we adopt three simple strategies to achieve them. (1) using larger kernel convolutions to enlarge the receptive field; (2) Introducing a multi-branch structure to obtain multi-scale information; (3) Introducing self-multiplication to achieve adaptability. After the above three improvements, our method outperforms the previous method by a large margin.
> * Advantages: Compared with the self-attention mechanism, we believe that our method has the following advantages：
>   * lower complexity: self-attention has quadratic complexity, which limits its applications for processing high-resolution segmentation images such as 2,048 x 1,024 images in cityscapes dataset. For our MSCA, it has linear complexity, which is more suitable for semantic segmentation task. In our paper, figure1(left)  clearly demonstrates the advantages when dealing with high-resolution images.
>   * Multi-scale Information aggregation:  For semantic segmentation task, it requires to process multi objects with various scale at the same time. Thus, it is critical to achieve multi-scale Information aggregation in this case. The self-attention mechanism ignores this, while our MSCA takes it into account.
>   * Channel attention: Channel attention has been proven important for vision tasks[2],[3]. For self-attention, it only considers the spatial attention. For our MSCA, it considers the self-adaptive property in  both channel and spatial  dimensions and achieves spatial and channel attention[3],[4] in a simple yet effective way.
> * Visualization: We adopt Grad-CAM[5] to conduct visualization to prove the effectiveness. Due to the limitation of openreview, we add visualization results in the supplementary material. From the visualization results, we can easily find two shortcomings of ConvNext: Insufficient receptive field and lack of multi-scale information.
>
> * Experimental results: SegFormer[6] and SETR[7] are common transformer-based models. Due to the above superiority, SegNeXt significantly surpasses them in various datasets including ADE20K, Cityscapes and COCO-Stuff, which is shown in our Table 8.
>
>
> [1]:  Liu, Z., Mao, H., Wu, C. Y., Feichtenhofer, C., Darrell, T., & Xie, S. (2022). A convnet for the 2020s. In *Proceedings of the IEEE/CVF Conference on Computer Vision and Pattern Recognition* (pp. 11976-11986).
>
> [2]: Hu, J., Shen, L., & Sun, G. (2018). Squeeze-and-excitation networks. In *Proceedings of the IEEE conference on computer vision and pattern recognition* (pp. 7132-7141).
>
> [3]: Woo, S., Park, J., Lee, J. Y., & Kweon, I. S. (2018). Cbam: Convolutional block attention module. In *Proceedings of the European conference on computer vision (ECCV)* (pp. 3-19).
>
> [4]: Wang, F., Jiang, M., Qian, C., Yang, S., Li, C., Zhang, H., ... & Tang, X. (2017). Residual attention network for image classification. In *Proceedings of the IEEE conference on computer vision and pattern recognition* (pp. 3156-3164).
>
> [5]: Selvaraju, R. R., Cogswell, M., Das, A., Vedantam, R., Parikh, D., & Batra, D. (2017). Grad-cam: Visual explanations from deep networks via gradient-based localization. In *Proceedings of the IEEE international conference on computer vision* (pp. 618-626).
>
> [6]:  Xie, E., Wang, W., Yu, Z., Anandkumar, A., Alvarez, J. M., & Luo, P. (2021). SegFormer: Simple and efficient design for semantic segmentation with transformers. *Advances in Neural Information Processing Systems*, *34*, 12077-12090.
>
> [7]:  Zheng, S., Lu, J., Zhao, H., Zhu, X., Luo, Z., Wang, Y., ... & Zhang, L. (2021). Rethinking semantic segmentation from a sequence-to-sequence perspective with transformers. In *Proceedings of the IEEE/CVF conference on computer vision and pattern recognition* (pp. 6881-6890).

---

> > ### Comment · Reviewer_r2Bq · 2022-08-05
> > **The rebuttal addresses most of my concerns**
> >
> > In the rebuttal stage, the authors address most of my concerns. I approve of designing a simple yet effective network for the segmentation task in this paper. Hence, I raise my score.

---

> > > ### Author Response · Authors · 2022-08-05
> > > **Thank you for raising rating.**
> > >
> > > Thanks for approving our design and rising your rating.

---

### Official Review · Reviewer_mij9 · 2022-07-11

**Rating:** 7
**Confidence:** 3
**Soundness:** 3 good
**Presentation:** 4 excellent
**Contribution:** 3 good

**Summary:**

This paper presents SegNeXt, a new convolution-based model for semantic segmentation. The authors first propose 4 properties that constitute a successful semantic segmentation model: 1) A strong encoder backbone, 2) Multi-scale information interaction, 3) Spatial attention, and 4) low computational complexity. Based on these requirements, the authors propose a convolutional encoder based on a new multi-scale convolutional attention module (MSCA). This module uses multi-branch convolutions of different sizes to generate an attention mask, which is then used to weight the input (via element-wise multiplication) to generate the output of the module.

As a decoder, the authors propose to concatenate the convolutional features from the last input stages and feed them as input to a Hamburger module.

The authors evaluate their proposed architecture on several datasets like ADE20K, Cityscapes, Pascal VOC, Pascal context, COCO-Stuff and iSAID. The results of the experiments show that SegNeXt matches or outperforms other models with similar number of parameters while requiring less computation.

**Questions:**

1) Why does the model need to use spatial attention? What is the property of spatial attention that makes it a key component for semantic segmentation models? (Be it self-attention or computed attention masks like in this paper)
2) In Section 1, Table 1, the authors mention that “Strong encoder denotes strong backbones, and adopts the advanced training strategy”. What is the “advanced training strategy”? Does this paper use it?

**Limitations:**

Yes

**Strengths And Weaknesses:**

## Strengths
+ The paper is well written and the new concepts and ideas are explained well and with enough detail. The paper is well structured and the content is presented in a clear and logical way.

+ The paper proposes an extensive evaluation to show the performance of the proposed model and how it compares against previous state-of-the art approaches. Additionally, the authors perform some ablation experiments to backup their choices of multi-scale in the MSCA module, Hamburger as a decoder and the decoder structure.

+ The proposed model outperforms previous state of the art approaches while also requiring less computation.

## Weaknesses
- A weakness of this paper is that it doesn’t justify enough the four properties argued to be necessary for a successful semantic segmentation model. The authors mention that by looking at previous work (DeepLabV3+, HRNet, SETR and SegFormer) they are able to conclude that a good model needs these four properties (strong backbone, multi-scale, attention and low complexity), but that might just indicate recent trends, not necessarily important properties. For example, a stronger justification to claim that the model needs to use spatial attention would be to design an experiment (or a citation to another paper) that shows why this property is needed, instead of assuming that because recent papers used attention, this property becomes necessary.

- The authors explicitly mention that
> “taking the aforementioned **analysis** into account, … we propose an efficient yet effective encoder-decoder architecture of semantic segmentation”.

  However, as I mentioned before, a more detailed analysis of the different properties of recent models would make the claims more significant.

- Just so the authors are aware, there is already another model called SegNext [1].

To summarize, the paper proposes a novel architecture for semantic segmentation which outperforms previous approaches. The authors perform a thorough evalution to empirically show the superior performance. However, some of the justifications are a bit weak or not supported enough with data.

[1] T. Forbes and C. Poullis, "Deep Autoencoders with Aggregated Residual Transformations for Urban Reconstruction from Remote Sensing Data," 2018 15th Conference on Computer and Robot Vision (CRV), 2018, pp. 23-30, doi: 10.1109/CRV.2018.00014.

---

> ### Author Response · Authors · 2022-08-02
> **Response to Reviewer mij9**
>
> Dear Reviewer mij9：
>
> Thanks for your effort for reviewing our paper and giving some kind suggestions. We are happy to see your positive comments and score. We hope the following responses could solve your concerns.
>
> * For Q1: "Why does the model need to use spatial attention? "
>
>   We believe that the attention mechanism is effective for semantic segmentation mainly because of two aspects.
>
>   * Firstly, attention mechanism can make the network focus on critical features and ignore noisy features automatically, which is an important property for vision tasks such as semantic segmentation [1]. It is also a key part of the human visual system [2],[3].
>   * The second point is our own understanding for attention mechanism. Attention mechanism is a dynamic process, which means it can adjust its output features according to its input features. Compared with traditional convolutional neural networks, which is a static process, it demonstrates better transfer learning ability [4],[5].  For example, We have trained a model on the ImageNet dataset, and we are going to transfer it to downstream tasks. During transfer learning, the network needs to process lots of unseen objects/scenes in ImageNet. For a static process, we learn the capability to deal with seen objects, which can not handle unseen objects/scenes well. For a dynamic process, we can learn the self-adaptive capability. It means the network can adjust output features based on input features. This allows the network to quickly adjust output when it deals with unseen objects, which is an important property for transfer ImageNet pre-trained weight to new benchmarks.
>
> * For Q2: "What is the “advanced training strategy”? Does this paper use it?"
>
>   The advanced training strategy is proposed by DeiT[6], which mainly contains more data augmentation, longer training epochs and strong regularization. It can improve the performance of the network on ImageNet and downstream tasks. Of course, we adopted it when training models on ImageNet.
>
>
>
> [1]:  Xu, K., Ba, J., Kiros, R., Cho, K., Courville, A., Salakhudinov, R., ... & Bengio, Y. (2015, June). Show, attend and tell: Neural image caption generation with visual attention. In *International conference on machine learning* (pp. 2048-2057). PMLR.
>
> [2]: Rensink, R. A. (2000). The dynamic representation of scenes. *Visual cognition*, *7*(1-3), 17-42.
>
> [3]: Corbetta, M., & Shulman, G. L. (2002). Control of goal-directed and stimulus-driven attention in the brain. *Nature reviews neuroscience*, *3*(3), 201-215.
>
> [4]: Guo, M. H., Xu, T. X., Liu, J. J., Liu, Z. N., Jiang, P. T., Mu, T. J., ... & Hu, S. M. (2022). Attention mechanisms in computer vision: A survey. *Computational Visual Media*, 1-38.
>
> [5]:  Wang, K., Gao, X., Zhao, Y., Li, X., Dou, D., & Xu, C. Z. (2019, September). Pay attention to features, transfer learn faster CNNs. In *International conference on learning representations*.
>
> [6]: Touvron, H., Cord, M., Douze, M., Massa, F., Sablayrolles, A., & Jégou, H. (2021, July). Training data-efficient image transformers & distillation through attention. In *International Conference on Machine Learning* (pp. 10347-10357). PMLR.

---

### Official Review · Reviewer_rY1T · 2022-07-11

**Rating:** 6
**Confidence:** 4
**Soundness:** 2 fair
**Presentation:** 3 good
**Contribution:** 2 fair

**Summary:**

This paper proposes a novel convolution-based architecture that could utilize convolutional attention in an effective way to encode contextual information for the segmentation task. The main contributions of this paper are (1) a new tailored network architecture (SegNeXt) that envokes spatial attention via multi-scale convolution features, (2) the illustration that the encoder with simple and cheap convolutions can exhibit improved performance than the vision transformer, and (3) the proposed network exhibit the state-of-the-art semantic segmentation performance.

**Questions:**

* Questions & Discussions
1. The novelty and the contribution of this paper should be more discussed. In recent years, the new methods related to vision transformer and ConvNext are popularly studied, and they showed extremely improved accuracy in terms of classification and segmentation. At the first gland, this paper just addresses the application of the ConvNext to the segmentation task. At this point, the reviewer is curious that the combination of the ConvNext to the segmentation task could be a novel contribution to this Neurips society.

2. Effectiveness of the proposed operation. The authors addressed in the abstract that “convolutional attention is a more efficient and effective way to encode contextual information than the self-attention mechanism in transformers”. However, the manuscript could not effectively exhibit this property which is the main contribution of this paper. The mathematical proof/illustration or experimental analysis for contextual information should be discussed to improve the quality of the manuscript.


**Ethics Review Area:**

["I don’t know"]

**Limitations:**

1. Major issues
- The limited novelty should be discussed. As illustrated above (Question), the proposed operation exhibits limited novelty. The reviewer expects that the limitations could be improved in the discussion periods.

- The experiments should be improved. As the authors commented in the contribution 2, the proposed module can perform better than vision transformers, “especially when processing object details”. As the reviewer already understand, the evaluation metric of “mean Intersection over Union (mIoU)” can qualitatively measure the object details. However, recent studies [1-3] proposed new evaluation metrics to measure the object detail (especially boundaries of the target objects) quantitatively. Otherwise, the visualization of the feature map could illustrate the novel feature extraction when processing object details. Please refer the activation maps [4]. To clear the authors’ addresses, more experimental or mathematical evidence should be justified.


[1] Fernandez-Moral, Eduardo, et al. "A new metric for evaluating semantic segmentation: leveraging global and contour accuracy." 2018 IEEE intelligent vehicles symposium (iv). IEEE, 2018.

[2] Lee, Kyungsu, et al. "Boundary-oriented binary building segmentation model with two scheme learning for aerial images." IEEE Transactions on Geoscience and Remote Sensing 60 (2021): 1-17.

[3] Cheng, Bowen, et al. "Boundary IoU: Improving object-centric image segmentation evaluation." Proceedings of the IEEE/CVF Conference on Computer Vision and Pattern Recognition. 2021.

[4] Selvaraju, Ramprasaath R., et al. "Grad-cam: Visual explanations from deep networks via gradient-based localization." Proceedings of the IEEE international conference on computer vision. 2017.


2. Minor issues
- Definition of the contextual information

- The detailed description should be improved. For instance, what does the arrow indicate in figure 3? The reviewer recommends authors to review the figures to improve the explanation details for all readers.

- The reviewer recommends reviewing the grammar and typo errors to improve the quality of the manuscript.


**Strengths And Weaknesses:**


1. Strengths
- The reviewer significantly understands the significance and importance of the task proposed in this paper. Segmenting various objects in real-world images is significantly important for many applications. Recent studies have researched the Vision-Transformer to improve the segmentation performance in vision fields, but it exhibits many limitations such as requirements a large number of images and datasets, and even heavy cost. However, this paper address that still the convolution and attention-based operations could exhibit significantly improved performance at a low cost.

- Reproducibility of the manuscript. The manuscript is well organized in terms of exhibiting the hyper-parameters and model architecture.

- The manuscript is well organized and well written.

2. Weakness
- More detailed descriptions are illustrated in the “Question” and “Limitation” sections. Please see below.

---

> ### Author Response · Authors · 2022-08-02
> **Response to Reviewer rY1T (1/2)**
>
> Thanks for your effort for reviewing our paper and giving some kind suggestions. We are happy to be affirmed in performance, reproducibility and writing.  We hope the following responses could solve your concerns.
>
> * For Question1: "The novelty and the contribution of this paper."
>
>   This paper is not a simple combination of ConvNeXt[1] and segmentation task. We do not deny we learn something from ConvNeXt, which is included in our reference list. Here,  we illustrate the difference between SegNeXt and ConvNeXt from three aspects: analysis, visualization and experimental results.
>
>   * Analysis:  The goal of this paper is to find simple and suitable operation / network for segmentation task. To achieve the target, we analyze and summarize the expected properties of segmentation task such as multi-scale information, strong encoder, attention mechanism (dynamic process) and low complexity. For ConvNeXt, it only satisfies the strong encoder and low complexity, but ignores the multi-scale information and attention mechanism, which are critical for segmentation task.
>
>   * Visualization: We adopt Grad-CAM[2] to conduct visualization to prove the effectiveness. Due to the limitation of openreview, we add visualization results in the supplementary material. From the visualization results, we can easily find two shortcomings of ConvNext: Insufficient receptive field and lack of multi-scale information. Meanwhile, our method makes up for its shortcomings.
>
>   * Experimental results: We replace our backbone with ConvNeXt and fairly compare ConvNeXt with our MSCAN on ADE20K dataset.  The results is shown in following table. Experiments also indicate the superiority of our method and confirm our view.
>
>     | Method              | Params.(M) | mIoU(SS) | mIoU(MS) |
>     | ------------------- | ---------- | -------- | -------- |
>     | SegNeXt w/ ConvNeXt | 28.4       | 43.2     | 44.5     |
>     | SegNeXt w/ MSCAN    | 27.6       | 48.5     | 49.9     |
>     | SegNeXt w/ ConvNeXt | 50.0       | 46.2     | 47.6     |
>     | SegNeXt w/ MSCAN    | 48.9       | 51.0     | 52.1     |
>
>
>
> * For Question2: "Effectiveness of the proposed operation."
>
>   In this paper, we compare our method with common transformer-based method such as SegFormer[3] and SETR[4]. As summarized in Table 1, transformer-based methods lack of multi-scale information and have a high computational complexity. These two drawbacks cause two problems: low performance and high computation.
>
>   * On the one hand, lacking of multi-scale information causes  performance reduction. It is demonstrated in Table 8, which SegNeXt outperforms SegFormer and SETR significantly.
>   * On the other hand, quadratic complexity causes high computing cost especially for high resolution images. As shown in Figure 1 left, SegNeXt significantly surpasses SegFormer when processing 2,048 x 1,024 images in cityscapes dataset.
>
> To be continued.

---

> > ### Author Response · Authors · 2022-08-02
> > **Response to Reviewer rY1T (2/2)**
> >
> > * For Question3: "The experiments should be improved."
> >
> >   We improve the experiments according to the comments.
> >
> >   * We adopt boundary-oriented intersection over union (B-IoU)[5] as evaluation metric to compare our method with SegFormer on cityscapes dataset. As shown in following Table, we also achieve better performance than SegFormer[1], which is a common transformer-based method.
> >
> >     | Method       | Params.(M) | GFLOPs | BIoU (SS) |
> >     | ------------ | ---------- | ------ | --------- |
> >     | SegFormer-B0 | 3.8        | 126.6  | 20.1      |
> >     | SegNeXt-T    | 4.3        | 50.5   | 21.3      |
> >     | SegFormer-B1 | 13.7       | 243.7  | 21.7      |
> >     | SegNeXt-S    | 13.9       | 124.6  | 22.7      |
> >     | SegFormer-B2 | 27.5       | 717.7  | 23.3      |
> >     | SegNeXt-B    | 27.6       | 275.7  | 24.2      |
> >     | SegFormer-B3 | 47.3       | 962.9  | 23.6      |
> >     | SegNeXt-L    | 48.9       | 577.5  | 25.1      |
> >
> >
> >
> >   * As shown in the response for Question1, visualization results based on Grad-CAM[2] are added in the supplementary material.
> >
> > * For Question4 "about paper writing":
> >
> >   We will check our writing about definition of different concept, detailed description about figures and tables, grammar and typo errors carefully for camera ready version.
> >
> > [1]: Liu, Z., Mao, H., Wu, C. Y., Feichtenhofer, C., Darrell, T., & Xie, S. (2022). A convnet for the 2020s. In *Proceedings of the IEEE/CVF Conference on Computer Vision and Pattern Recognition* (pp. 11976-11986).
> >
> > [2]: Selvaraju, R. R., Cogswell, M., Das, A., Vedantam, R., Parikh, D., & Batra, D. (2017). Grad-cam: Visual explanations from deep networks via gradient-based localization. In *Proceedings of the IEEE international conference on computer vision* (pp. 618-626).
> >
> > [3]: Xie, E., Wang, W., Yu, Z., Anandkumar, A., Alvarez, J. M., & Luo, P. (2021). SegFormer: Simple and efficient design for semantic segmentation with transformers. *Advances in Neural Information Processing Systems*, *34*, 12077-12090.
> >
> > [4]:  Zheng, S., Lu, J., Zhao, H., Zhu, X., Luo, Z., Wang, Y., ... & Zhang, L. (2021). Rethinking semantic segmentation from a sequence-to-sequence perspective with transformers. In *Proceedings of the IEEE/CVF conference on computer vision and pattern recognition* (pp. 6881-6890).
> >
> > [5]:  Lee, K., Kim, J. H., Lee, H., Park, J., Choi, J. P., & Hwang, J. Y. (2021). Boundary-oriented binary building segmentation model with two scheme learning for aerial images. *IEEE Transactions on Geoscience and Remote Sensing*, *60*, 1-17.

---

> > > ### Comment · Reviewer_rY1T · 2022-08-09
> > > **Response to Rebuttal**
> > >
> > > * I would like to express my appreciation to the authors to take the time to my questions. Most of my concerns were resolved in the rebuttal phase; mainly related to the novelty and experimental improvement. Therefore, I increase my rating from four to six through the discussion phase. In addition,  I still agree that this proposed operation is significantly similar to the previous study (ConvNext). However, experimental results could express the novelty of the proposed operator, I agree that the proposed operator could not be restricted due to its similarity to the previous study. The empirical analysis could be an alternative way to exhibit the novelty of the paper. My remaining concern is about the extensibility of the proposed network as a generalized operation for the improvement of CNN compared to transformers. This could be discussed in the Discussion section.
> > >
> > > * Furthermore, for reproducibility and the improvement of the deep learning society, I would strongly address that the code would be published in public. In hopeful expectation, I will adjust my rating in good faith.

---

> > > > ### Author Response · Authors · 2022-08-09
> > > > **Thanks for your response.**
> > > >
> > > > Thanks for your positive response and raising your rating.
> > > >
> > > >
> > > > We are sure that the extensibility of proposed operation will be discussed in the discussion section and the code will be public as mentioned in abstract.
> > > >
> > > > Sincerely,
> > > >
> > > > Authors

---

> > > > ### Author Response · Authors · 2022-08-10
> > > > **About open source**
> > > >
> > > > In fact, we had organized the related code like SegFormer[1], which can be released at any time.
> > > >
> > > > Due to the anonymity rules of NeurIPs 2022, we can not provide it now.
> > > >
> > > > Trust us, the code will be public as mentioned in abstract.
> > > >
> > > > [1]: Xie, E., Wang, W., Yu, Z., Anandkumar, A., Alvarez, J. M., & Luo, P. (2021). SegFormer: Simple and efficient design for semantic segmentation with transformers. Advances in Neural Information Processing Systems, 34, 12077-12090.

---

> ### Author Response · Authors · 2022-08-07
> **Would you give us a response ?**
>
> Dear Reviewer rY1T:
>
> We are sorry to bother you. Firstly, thanks for your effort for reviewing our paper and giving some valuable suggestions. We fully understand and appreciate reviewers‘ selfless contributions to the community. Similarly, we hope reviewers can understand paper authors.
>
> Our team has made great efforts and spent lots of time and resources to conduct qualitative and quantitative experiments, answer related questions, and refine paper in the rebuttal stage.
>
> Since the author-reviewer discussion is close to deadline, we hope we can get your response before the deadline. If you have other concerns, we are always willing to discuss.
>
> Sincerely,
>
> Authors

---

> > ### Author Response · Authors · 2022-08-09
> > **Could we have a discussion ?**
> >
> > Dear reviewer rY1T:
> >
> > We sincerely thank you for the review and comments. We have provided corresponding responses and results, which we believe have covered your concerns.
> >
> > As you mentioned in limitations, you expects that the limitations could be improved in the discussion periods. We also hope to further discuss with you to address the misunderstanding of our work. Now, it has been less than 12 hours to the end of the discussion, could we have a discussion during this period?
> >
> >
> > Best,
> >
> > Authors

---

### Meta-Review · Area_Chair_DG9i · 2022-08-25

**Recommendation:** Accept
**Confidence:** Less certain

**Metareview:**

Four knowledgeable referees reviewed this submission. The reviews raised concerns about the novelty of the proposed approach (rY1T, S4w5), the motivation of the model design and properties (mij9, r2Bq), and the empirical evidence to support some of the effectiveness and efficiency claims (rY1T, r2Bq, S4w5). The rebuttal addresses the reviewers' concerns by (1) highlighting the differences of the proposed approach with ConvNext, (2) providing additional comparisons with state-of-the-art methods as suggested by the reviewers, (3) performing ablations of the MSCA design which empirically emphasize its advantages, and (4) partially clarifying the motivation. The authors engage in discussion with the reviewers and provide additional clarifications (e.g. what will be introduced in the main body of the paper, and whether the code will be released). During the discussion phase, the reviewers show some hesitations wrt novelty of the proposed approach which is perceived as incremental wrt ConvNext. However, the reviewers agree that the paper is well written, the approach is simple and appears effective, and the experimental evidence is extensive and supports the claims made in the manuscript. The reviewers appreciate the benchmarking efforts of this work and lean towards acceptance. The AC agrees with the reviewers' assessment that the strength of this paper lies in its extensive experimental validation, and recommends to accept.

**Award:**

No

---

### Decision · Program_Chairs · 2022-09-14

Accept